# Evaluation of Country Dietary Habits Using Machine Learning Techniques in Relation to Deaths from COVID-19

**DOI:** 10.3390/healthcare8040371

**Published:** 2020-09-29

**Authors:** María Teresa García-Ordás, Natalia Arias, Carmen Benavides, Oscar García-Olalla, José Alberto Benítez-Andrades

**Affiliations:** 1SECOMUCI Research Group, Escuela de Ingenierías Industrial e Informática, Universidad de León, Campus de Vegazana s/n, C.P., 24071 León, Spain; mgaro@unileon.es; 2SALBIS Research Group, Department of Nursing and Physiotherapy Health Science School, University of León, Avenida Astorga s/n, Ponferrada, 24401 León, Spain; narir@unileon.es; 3SALBIS Research Group, Department of Electric, Systems and Automatics Engineering, University of León, Campus of Vegazana s/n, León, 24071 León, Spain; carmen.benavides@unileon.es; 4Artificial Intelligence Department, Xeridia S.L., Av. Padre Isla 16, 24002 León, Spain; oscar.olalla@xeridia.com

**Keywords:** COVID-19, countries, fat, protein, KCal, deaths, machine learning, K-Means

## Abstract

COVID-19 disease has affected almost every country in the world. The large number of infected people and the different mortality rates between countries has given rise to many hypotheses about the key points that make the virus so lethal in some places. In this study, the eating habits of 170 countries were evaluated in order to find correlations between these habits and mortality rates caused by COVID-19 using machine learning techniques that group the countries together according to the different distribution of fat, energy, and protein across 23 different types of food, as well as the amount ingested in kilograms. Results shown how obesity and the high consumption of fats appear in countries with the highest death rates, whereas countries with a lower rate have a higher level of cereal consumption accompanied by a lower total average intake of kilocalories.

## 1. Introduction

Many pneumonia cases of unknown cause emerged in Wuhan, Hubei, China in December 2019. Deep sequencing analysis from lower respiratory tract samples indicated a previously unknown coronavirus, which was named SARS-CoV-2 [1]. COVID-19, caused by SARS-CoV-2, was first reported in Wuhan but it quickly spread throughout the world becoming a global public health emergency [2].

It is transmitted by direct contact with respiratory drops that are emitted through a sick person’s cough or sneeze [3,4]. Its contagiousness depends on the amount of the virus in the airways.

These drops infect another person through the nose, eyes, or mouth directly but they can also infect by touching the nose, eyes, or mouth with hands that have previously touched surfaces contaminated by these drops [5].

Kampf et al. [3] conducted a study in which they revealed that coronaviruses can remain infectious on inanimate surfaces for up to 9 days. However, surface disinfection with 0.1% sodium hypochlorite or 62–71% ethanol significantly reduces coronavirus infectivity on surfaces within 1 min from exposure time.

Transmission by air over distances greater than 2 m seems unlikely.

Most people get COVID-19 from other people with symptoms. However, there is increasing evidence of the role that people have in the transmission of the virus before the development of symptoms or with mild symptoms [6,7].

Chen et al. [8] carried out a descriptive study of the epidemiological and clinical characteristics of 99 cases of COVID-19 in Wuhan. The symptoms found were as follows; fever (83%), cough (82%), shortness of breath (31%), muscle ache (11%), confusion (9%), headache (8%), sore throat (5%), rhinorrhea (4%), chest pain (2%), diarrhea (2%), and nausea and vomiting (1%). According to imaging examination, 75% of the patients showed bilateral pneumonia, 14% showed multiple mottling and ground-glass opacity, and 1% had pneumothorax.

At present, there is no vaccine or antiviral treatment for Covid19. For the moment, isolation and supportive care including oxygen therapy, fluid management, and the administration of antimicrobials to alleviate the symptoms and prevent organ dysfunction are the measures that are being taken.

A recent study [9] in 18 rhesus macaques demonstrated that Remdesivir (GS-5734) provided a clear clinical benefit, with a reduction in clinical signs, reduced virus replication in the lungs, and decreased presence and severity of lung lesions. Furthermore, Liu et al. [10] found that up to 10 commercial medicines that may form hydrogen bounds to key residues within the binding pocket of COVID-19 could have a higher mutation tolerance than lopinavir or ritonavir.

According to Sarma et al., after conducting a systematic review, they found that hydroxychloroquine together with azithromycin may be a rather promising combination for combating COVID-19 [11]. Therefore, at the moment, there is no cure for COVID-19 but there are ongoing efforts in the development of a vaccine [12].

As in all disciplines, machine learning, deep learning, and artificial intelligence techniques can be useful in providing new information about the still unknown COVID-19. Gozes et al. [13] demonstrate that non-contrast thoracic CT images is an effective tool in detection, quantification, and follow-up of the disease. The authors used U-net for the segmentation of the lung region step [14] followed by a Resnet-50-2D deep convolutional neural network architecture [15] to classify. The authors achieved classification results for Coronavirus vs non-coronavirus cases per thoracic CT studies of 0.996 AUC (95%CI). The system also provides measurements on the progression of patients over time.

Chest CT are also used in [16]. Li et al. developed a deep learning model (COVNet) to extract visual features from volumetric chest CT exams for the detection of COVID-19. The developed model can accurately detect COVID-19 and also differentiate it from pneumonia and other lung diseases with really promising results.

Artificial intelligence techniques and regression analysis have also been used in [17]. This work shows the impact of weather parameters on confirmed cases of COVID-19, and it has demonstrated that the relative humidity and maximum daily temperature had the highest impact on the confirmed cases. The relative humidity in the main case study, with an average of 77.9%, affected the confirmed cases positively and maximum daily temperature, with an average of 15.4 ${}^{\circ }$C, affected the confirmed cases negatively.

In this study, artificial intelligence has been used to reveal very interesting information about the relationship between COVID-19 and the dietary habits in different countries all around the world. It is well known that food affects health and diseases [18,19].

Furthermore, recent researches have been proved that obesity increases the risk of adverse outcomes of COVID-19 and even it is highly associated with its mortality [20,21].

In this work, it has been discovered that dietary habits are related to COVID-19 mortality, so taking care of diet can be a good way to prevent COVID-19 death risk.

Therefore, the aim of this research is to identify patterns that make it possible to put the focus of attention on countries that today are in the early stages of the virus’s expansion but share nutritional characteristics with the countries that have suffered the most from this pandemic and may represent a danger in the future.

## 2. Methods

### 2.1. Principal Component Analysis (Pca)

Principal component analysis (PCA) [22] is one of the most familiar methods of multivariate analysis which uses the spectral decomposition of a correlation coefficient or covariance matrix. In other words, PCA is a procedure for reducing the dimensionality of the variable space by representing it with a few orthogonal (uncorrelated) variables that capture most of its variability. Dimensions must be reduced when there are too many characteristics in a data set, making it difficult to distinguish between those that are relevant and those that are redundant or do not provide significant information.

PCA is a feature extraction technique, that is, the input variables are combined in a specific way so that the least important variables are discarded while preserving the most valuable parts of all the variables. PCA results are new features that are independent of each other.

The first step in calculating PCA is the data standardization so that each variable contributes equally to analysis following Equation (Equation 1),
(1)$$z=\frac{x-\mu }{\sigma }$$
where *x* is the original data, μ is the mean, and σ is the standard deviation.

After that, the covariance matrix must be computed. The covariance between two variables *X* and *Y* is computed following Equation (Equation 2),
(2)$$cov\left(X,Y\right)=\frac{1}{n-1}\sum_{i=1}^{n}\left(X_{i}-\bar{x}\right)\left(Y_{i}-\bar{y}\right)$$
with *n* being the number of data, $X_{i}$ and $Y_{i}$ the current data, and $\bar{x}$ and $\bar{y}$ the mean. Computing all of these values, a square matrix of n×n is obtained. This matrix is called the covariance matrix.

The next step is compute the eigenvectors and their corresponding eigenvalues. The eigenvector of the covariance matrix is the vector which satisfies Equation (Equation 3).
(3)$$A\bar{v}=\lambda \bar{v}$$

In this case, *A* is the covariance matrix, $\bar{v}$ is the eigenvector, and λ is a scalar value called the eigenvalue.

Once all of the eigenvectors and the corresponding eigenvalues are computed, the *k* eigenvectors with the largest eigenvalues are selected. This parameter “k” is the dimension of the new dataset. The last step is to project the data points in accordance with the new axes.

### 2.2. K-Means

The K-means clustering method is used to find patterns or similarity between data. The first step is determine the number of centers or the number of groups *k*. These centers (centroids) are randomly initialized.

Each data point is assigned to the closest centroid, and each collection of points assigned to a centroid represents a cluster.

After this first allocation step, the centroid of each cluster is updated taking into account the data points assigned to it. This process is repeated until the data points remains constant in the same cluster or until the centroids remain the same.

### 2.3. Clustering Metric: Davies–Bouldin

To obtain the appropriate number of groupings that we have to make between the countries, the Davies–Bouldin clustering metric [23] has been used. The Davies–Bouldin metric is defined as the mean value among all the clusters of the samples $M_{k}$ (see Equation (Equation 4)).
(4)$$DB=\frac{1}{K}\sum_{k=1}^{K}M_{k}$$This expression is equivalent to Equation (Equation 5):(5)$$DB=\frac{1}{K}\sum_{k=1}^{K}max_{k^{\prime }\neq k}\left(\frac{\delta_{k}+\delta_{k^{\prime }}}{{△}_{kk^{\prime }}}\right)$$
with $\delta_{k}$ being the mean distance of the points belonging to cluster $C_{k}$ to their barycenter $G_{k}$, and ${△}_{kk^{\prime }}$ the distance between barycenters $G^{k}$ and $G^{k^{\prime }}$ (see Equation (Equation 6)).
(6)$${△}_{kk^{\prime }}=d\left(G^{k},G^{k^{\prime }}\right)=\left|\right|G^{k}-G^{k^{\prime }}\left|\right|$$

## 3. Experiments and Results

### 3.1. Dataset

The COVID-19 Healthy Diet Dataset [24] has been used in this work in order to study the relationship between the diet of the different countries and the number of deaths caused by the disease.

The COVID-19 Healthy Diet Dataset combines data of different types of food and COVID-19 cases and deaths all around the world. The dataset contains information about the following types of food for each of the 170 countries; alcohol, animal products, animal fats, aquatic products, cereals, eggs, seafood, fruits, meat, miscellaneous, milk, offal, oilcrops, pulses, spices, starchy roots, stimulants, sugar crops, sugar and sweeteners, treenuts, vegetal products, vegetable oils, and vegetables.

It is made up of four csv files containing.

Percentages of fat consumed from each type of food listed.Percentages of food supply (in kg) for each type of food listed.Percentages of energy (in kilocalories) consumed from each type of food listed.Percentages of protein consumed from each type of food listed.

Information about obesity, undernourished, confirmed cases, deaths, recovered, activity levels, and the population of each country is also included in the dataset.

Although race has been proved to be associated with mortality [25,26], our data does not contain information about race distribution by countries, so we have decided to exclusively take into account their dietary habits.

Furthermore, information about the consumption of kilocalories per country has been obtained from FAOSTAT [27] in order to cross reference this information with the COVID-19 Healthy Diet Dataset.

### 3.2. Experiments

As stated in the previous section, the data set is made up of 94 characteristics related to 23 types of food. To avoid using multiple features that provide the same information, a transformation of the data has been carried out using PCA. In this case, the number of features has been reduced to 23, which is the minimum number necessary to retain 95% of the information. With this, we not only prevent duplicate information from biasing the results of the study, but we also reduce the time necessary to manipulate them.

Once we have reduced the data, K-Means has been applied with the intention of grouping the 170 countries into clusters based on that reduced information of their food consumption. The intention of this group is to try to identify patterns that make it possible to put the focus of attention in countries that today are in the early stages of the virus’s expansion but share nutritional characteristics with the countries that have suffered the most from this pandemic and may represent a danger in the future.

In order to determine the best number of clusters, the Davies–Bouldin index has been used. In Figure 1, a graph with the Davies–Bouldin index can be seen. According to these results, we have chosen 20 clusters, as the benefit of adding additional clusters is minimal and there is a significant point and change in slope at that point in the graph.

The distribution of all the countries around the 20 clusters can be shown in Figure 2.

It is important to highlight that this grouping has been carried out by only taking into account the diet of each of the countries. Once the countries were grouped together, the average percentage of deaths from COVID-19 for each cluster was evaluated to try to identify the dietary patterns that influence the greatest number of deaths from detected infections. In Figure 3, the information related to the 3rd quartile for each of the clusters is shown.We have chosen this statistical information to have a more realistic view of the distribution of cases of deaths by cluster than that which would be obtained simply using the mean (where very high values can distort the measurement).

Taking this information into account, a threshold has been established and each of the clusters has been labeled as a cluster with a high probability or a cluster with a low probability of death. Clusters 3, 4 and 17 were labeled with a high probability of death, which includes, taking into account Figure 2, a total of 30 countries.

A study has been carried out of the main types of food that, according to the group carried out, most affect deaths from COVID-19. In Figure 4, the influence of animal products, milk, cereals, sugars and sweets, meat, and animal fats on the final result of the disease is clear. As we can see in all food categories, the high death group shows less variation in the data based on standard deviation statistics which implies a more cohesive cluster.

As we can see, these results are aligned with the importance of functional food that enrich the diet of Mediterranean countries [28].

Furthermore, a study of the amount of obesity and undernourished has been carried out (see Figure 5). As we can see, countries with a high percentage of obesity had a higher risk of death. In contrast, the undernourished percentage is lower for higher risk countries. These results can confirm that eating products like meat, animal fats, milk, or sweeteners increases the risk of death caused by COVID-19.

Moreover, an evaluation of the consumption of kilocalories has been carried out, reaching the interesting conclusion that the countries that belong to the high risk group consume 3277.5 Kcal per day on average while the rest of the countries consume 2764.3 Kcal on average. These results show a difference in caloric consumption in the countries with a population at risk of 18.57% compared to the rest of the countries (Figure 5).

According to our machine learning process, the high death risk cluster is made up of 30 countries. In Table 1, we can see all of them.

Fifteen out of 30 countries appear in the top 30 of countries with more deaths until May 2020. Our results could be interesting in establishing how the rest of the countries belonging to our high risk cluster could evolve in the future based on their food consumption habits if the virus is not controlled.

The source code of the full experimentation is shared as Appendix A to allow other researchers to replicate the entire process reliably.

## 4. Discussion and Conclusions

In this work, a study has been carried out on the mortality of people infected with the SARS-CoV-2 virus, taking into account the type of food in the country in which they live. For this purpose, 94 characteristics related to the amount of fat, protein, and energy (kilocalories), as well as the amount of food ingested in kilograms (Kg), from different food groups, have been used. Because of the possibility that many of these characteristics were highly correlated with others, a reduction in characteristics has been made using principal component analysis (PCA) so that 95% of the variance of the data set is maintained. This resulted in a 75.53% reduction in the data, leaving only 23 characteristics at the end.

A data pooling was then carried out using the well-known K-Means clustering technique. To determine the number of clusters, a study was carried out using the Davies–Bouldin [23,29] metric which indicated that the optimal number of clusters according to their characteristics was 20. The average number of countries per cluster is 8 with a standard deviation of 5.28.

By studying the average percentage of deaths per cluster, two groups of countries were created: “high deaths” and “normal deaths”. Analyzing the dietary patterns of the two groups of countries—high deaths and normal deaths—it was observed that the consumption of animal products, animal fats, milk, sweeteners, and meat in countries with a high risk of death was higher while the consumption of cereals was higher in those with a lower risk of death. In addition, it was observed that countries with a high degree of obese people and with a higher average daily caloric intake, are related to a higher risk of death from COVID-19 while countries with a high number of undernourished people do not show an increase in these percentages. Obesity has been observed in other studies as a risk factor, tripling the likelihood of severe condition from COVID-19 [30].

Finally, the current number of deaths per country has been checked and it has been detected that 50.00% of the countries that appear in the “high deaths” cluster are among the 30 countries with the most deaths. This leads us to consider countries at risk to those that belong to the cluster we have created for having a similar diet, among other possible factors related to the lifestyle of its population.

In future work, adding additional information by country such as race, age, or socioeconomic status could help to perform a better clustering of countries and, consequently, a more powerful study of the impact of the virus.

## Figures and Tables

**Figure 1 healthcare-08-00371-f001:**
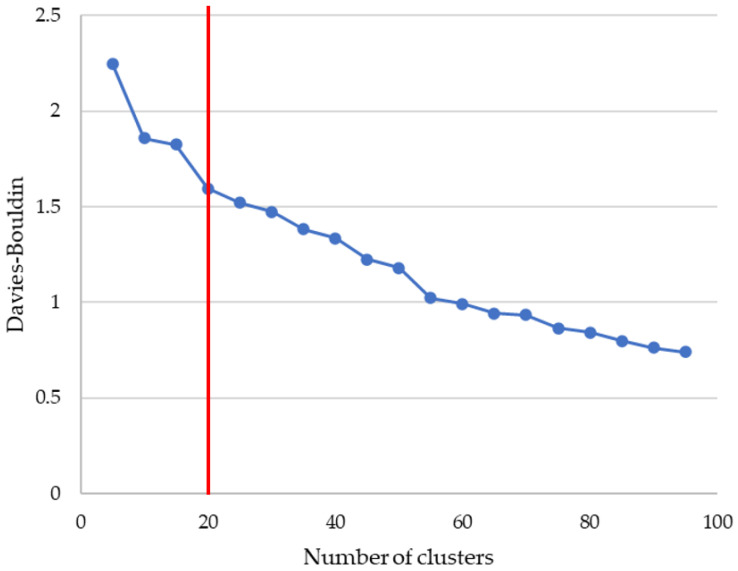
Study of the best number of clusters using Davies–Bouldin index.

**Figure 2 healthcare-08-00371-f002:**
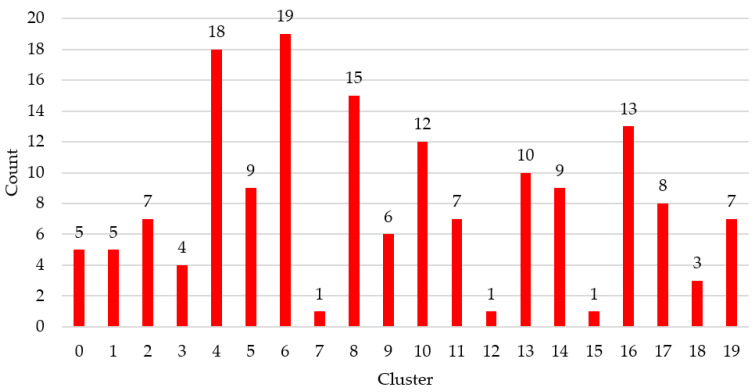
Number of countries per cluster.

**Figure 3 healthcare-08-00371-f003:**
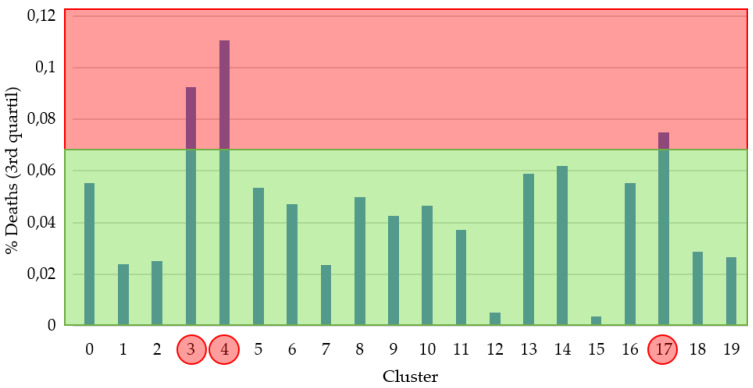
3rd quartile % of deaths per cluster.The red part represents a high % of deaths and the green part a low one. The red circles show the groups that fall into the red area.

**Figure 4 healthcare-08-00371-f004:**
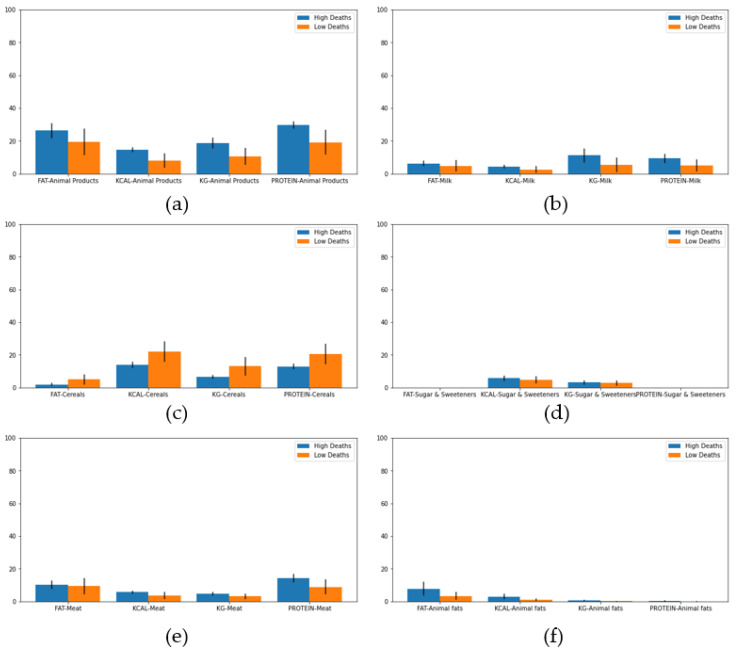
Mean percentage and standard deviation of consumption for different food in high and low death clusters. (**a**) Animal products, (**b**) Milk, (**c**) Cereals, (**d**) Sugar and Sweeteners, (**e**) Meat, and (**f**) Animal fats.

**Figure 5 healthcare-08-00371-f005:**
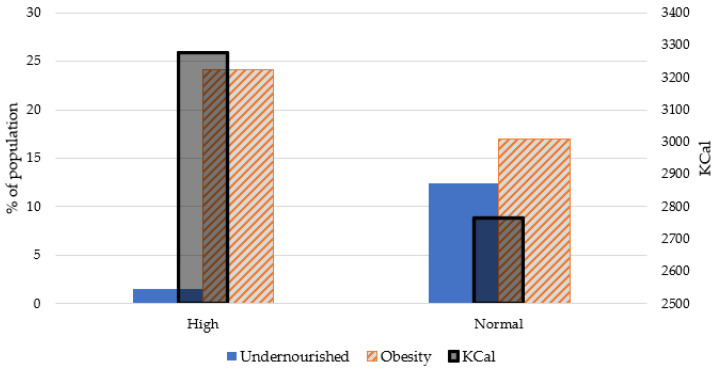
Obesity and undernourished diseases for high and normal death clusters.

**Table 1 healthcare-08-00371-t001:** Countries on high risk of death cluster. In green, we can see the countries that also appear in the Top 30 of countries with more deaths at the end of May 2020 based on COVID-19 Dashboard by the Center for Systems Science and Engineering (CSSE) at Johns Hopkins University.

Australia	Austria	Bahamas	Barbados	Belgium
Canada	Cyprus	Czechia	Denmark	France
Germany	Greece	Hungary	Ireland	Israel
Italy	Kazakhstan	Latvia	Lithuania	Netherlands
New Zealand	Norway	Poland	Portugal	Slovakia
Slovenia	Spain	Sweden	Switzerland	USA

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
