# Peer review of "Evaluation of Country Dietary Habits Using Machine Learning Techniques in Relation to Deaths from COVID-19"

_healthcare, 2020, doi:10.3390/healthcare8040371_

Round 1

Reviewer 1 Report

In this study, the authors conducted the eating habits associated with death from COVID-19. The surveillance of COVID-19 patients from 170 countries and their eating habits with 23 types of food and ingested kilograms, revealing high death rates with high consumption of fats while low death rates with high consumption of cereal by a lower average intake of kilocalories. The findings are important and somewhat interesting, although there have been many reports saying that increased morbidity and mortality of obese and overweight in COVID-19 patients. However, there are still some drawbacks to solve before final publication.

Major points:

(1) The distribution of 170 countries near cover all human races. However, it is found that race is associated with mortality among hospitalized COVID-19 patients. (Yehia et al., doi: 10.1001/jamanetworkopen.2020.18039; Booker et al., J Gerontol Nurs. 2020 Sep 1;46(9):4-6. doi: 10.3928/00989134-20200811-01.). How to exclude the factor when analyzed the pan-data of eating habits from COVID-19 patients? The authors need to explain or state.

(2) In Figure 4, are there any significant statistic differences between two groups? It may help to clarify the conclusion.

Minor points:

(1) Line 14, COVID-19 is not the name of the novel coronavirus. In fact, it is called SARS-CoV-2.

(2) Line 14-15, the sentence is misunderstanding. There are no evidences that SARS-CoV-2 originated in Wuhan. Rephrase it.

(3) It seems that the term “dietary habits” is more appropriate than “eating habits”.

Author Response

Reviewer#1 Comment#1 (C1.1)

In this study, the authors conducted the eating habits associated with death fromCOVID-19. The surveillance of COVID-19 patients from 170 countries and theireating habits with 23 types of food and ingested kilograms, revealing high deathrates with high consumption of fats while low death rates with high consumptionof cereal by a lower average intake of kilocalories. The findings are importantand somewhat interesting, although there have been many reports saying thatincreased morbidity and mortality of obese and overweight in COVID-19 patients.However, there are still some drawbacks to solve before final publication.

Authors – Answer#1 (A1.1)

Thank you very much for your feedback. We really appreciate your effort review-ing our paper and the rich comments which will lead in a better version of thepublication.

Reviewer#1 Comment#2 (C1.2)

The distribution of 170 countries near cover all human races. However, it isfound that race is associated with mortality among hospitalized COVID-19 pa-tients. (Yehia et al., doi: 10.1001/jamanetworkopen.2020.18039; Booker et al., JGerontol Nurs. 2020 Sep 1;46(9):4-6. doi: 10.3928/00989134-20200811-01.). Howto exclude the factor when analyzed the pan-data of eating habits from COVID-19 patients? The authors need to explain or state.

Authors – Answer#2 (A1.2)

Thank you for your comment. It is really good point. We have added these twopapers to our dataset section and we have explain why we can not deal with thatusing the current dataset:

Although race has been proved to be associated with mortality onCOVID-19 patients, our data does not contain information aboutrace distribution by countries so we have decided to exclusively takeinto account their dietary habitsFurthermore we have added this limitation in our conclusion sections.In future work, adding additional information by country suchas race, age, or socioeconomic status could help to perform a betterclustering of countries and, consequently, a more powerful study ofthe impact of the virus.

Reviewer#1 Comment#3 (C1.3)

In Figure 4, are there any significant statistic differences between two groups? Itmay help to clarify the conclusion.

Authors – Answer#3 (A1.3)

You are right. We have added information about statistic differences in theexperiments section:As we can see in all food categories, the high death group showsless variation in the data based on standard deviation statistics whichimplies a more cohesive cluster.Furthermore, we have modify the caption of the figure to indicate that the datashown is the mean and standard deviation of each cluster.

Reviewer#1 Comment#4 (C1.4)

Line 14, COVID-19 is not the name of the novel coronavirus. In fact, it is calledSARS-CoV-2.Authors – Answer#4 (A1.4)We fully agree with the reviewer on that point. We have modified it and we havealso indicate that COVID-19 is the disease caused by SARS-CoV-2.

Reviewer#1 Comment#5 (C1.5)Line 14-15, the sentence is misunderstanding. There are no evidences that SARS-CoV-2 originated in Wuhan. Rephrase it.

Authors – Answer#5 (A1.5)

We agree with the reviewer. We have indicated that the COVID-19 diseasewas first reported in Wuhan as indicated in the corresponding referenceChen,H.; Guo, J.; Wang, C.; Luo, F.; Yu, X.; Zhang, W.; Li, J.; Zhao, D.;Xu, D.; Gong, Q.; Liao, J.; Yang, H.;213Hou, W.; Zhang, Y. Clinical char-acteristics and intrauterine vertical transmission potential of COVID-19 infec-tion in nine pregnant women: a retrospective review of medical records.TheLancet2020,395, 809–815.215doi:10.1016/S0140-6736(20)30360-3, instead of theSARS-CoV-2 coronavirus which as you said, there is no evidence that it originatedin Wuhan. We have rewrite this sentence according to the previous comment toclarify it.

Reviewer#1 Comment#6 (C1.6)

It seems that the term “dietary habits” is more appropriate than “eating habits”.

Authors – Answer#6 (A1.6)

We think that the term proposed by the reviewer is clearly better than the oneemployed in the paper. We have reviewed the document and have changed itentirely.

Reviewer 2 Report

The topic is interesting from the computer science perspective and from health care perspective, the paper is a good technical paper. The paper offers an appropriate research method. I really see findings of novelty in the paper. The author employed a well known but interesting ML analysis for a hot topic .

It is clear, in my opinion that the paper contributes to the body of knowledge. The subject is of topical interest and importance owing to its relation to current events.

In my opinion, introduction section should be improved, focusing on those data relevant for the paper. Table 1 caption should contain explanation about the used coloring.

A discussion of the limits of your work and its advantages/comparison/positioning in relation to existing similar works would be also appreciated.

I think the references are sufficient.

Author Response

Reviewer #2 Comment#1 (C2.1)

The topic is interesting from the computer science perspective and from health care perspective, the paper is a good technical paper. The paper o_ers an appro-priate research method. I really see _ndings of novelty in the paper. The author employed a well known but interesting ML analysis for a hot topic .

It is clear, in my opinion that the paper contributes to the body of knowledge. The subject is of topical interest and importance owing to its relation to current events.

Authors (Answer#1 (A2.1)

We thank the reviewer for the time it took him to review this document and the good evaluations received.

Reviewer #2 Comment#2 (C2.2)

In my opinion, introduction section should be improved, focusing on those data relevant for the paper. Table 1 caption should contain explanation about the used coloring.

Authors { Answer#2 (A2.2)

We agree with the reviewer and we have added some new references connected with the relationship between obesity and COVID-19. The following lines have been included in the introduction section: Furthermore, recent researches have been proved that obesity in-creases the risk of adverse outcomes of COVID-19 and even it is highly associated with its mortality [20,21]. In this work, it has been discovered that dietary habits are related to COVID-19 mortality so taking care of diet can be a good way to prevent COVID-19 death risk. Also information about the green cells is indicated in the caption of Table 1: In green, we can see the countries that also appear in the Top 30 of countries with more deaths at the end of May 2020 based on COVID- 19 Dashboard by the Center for Systems Science and Engineering (CSSE) at Johns Hopkins University.

Reviewer #2 Comment#3 (C2.3)

A discussion of the limits of your work and its advantages/comparison/positioning in relation to existing similar works would be also appreciated.

Authors { Answer#3 (A2.3)

We agree with the reviewer in the lack of a discussion of the limits of our work. We have added it in the conclusion sections: In future work, adding additional information by country such as race, age, or socioeconomic status could help to perform a better clustering of countries and, consequently, a more powerful study of the impact of the virus. We have not found any work similar to ours which allows us to perform a compar-ison. In our work, we have evaluated all the countries through a machine learning clustering based on the dietary habits. Other researchers found a connection be-tween obesity and mortality but they do not take into account the dietary habitsof similar countries to carried out a risk prediction.

Reviewer 3 Report

Summary:

In this paper, the authors use the K-means clustering method, an unsupervised machine learning algorithm, to group countries in clusters according to the diet of each country. Then they evaluated the average percentage of deaths per cluster and by analysing the data obtained they made several conclusions.

Main remarks:

Overall, the manuscript contains nice contributions. However, some minor issue should be addressed.

  • The title can lead to a false assumption. “Evaluation of Eating Habits in Relation to Deaths from COVID-19 Using Machine Learning Techniques” leads the reader to think that you used machine learning techniques to relate the eating habits with the death of COVID-19. You surely relate the two but not using machine learning techniques since you only used K-means clustering to group countries according to the diet of each country. In the main text page 4 line 140 you state this important information. So, I would slightly alter the title to not mislead the reader.
  • Page 3 line 84: the mu and sigma are indicated as being the mean and standard deviation, but there is no indication of what. I suppose it is the mean and the standard deviation of the population.
  • Page 3 line 85: the same comment as my previous one. I think it would be clearer if it is mentioned to what the mean refers to.
  • Page 3 equation 3: I think it would be clearer if there is a mention of which letter is used for the eigenvector.
  • Page 4 lines 141 and 142: It states “the percentage of deaths from COVID-19 for each cluster was evaluated” but there is no indication of how it was evaluated. This indication only appears on page 7 line 181: “the average percentage of deaths per cluster”. I would be clearer if this indication appeared in this line.
  • Page 6 line 166: It states “12 out of 30 countries appear in the top 30 of countries with more deaths…” but in Table 1 of page 7 there are 15 countries in green which “appear in the top 30 of countries with more deaths” and in page 7 line 191 it is referred that “46,67% of the countries that appear in the high deaths cluster are among the 30 countries with the most deaths”. This 46,67% correspond to 14 countries. So, it is not clear which is the real number of countries that appear in the top 30 of countries with more deaths.
  • It would be a must to provide the code/scripts used in the analysis to make this work truly reproducible and to validate the results. The authors should add the code/script in the supplementary materials.

Author Response

Reviewer #3 Comment#1 (C3.1)

In this paper, the authors use the K-means clustering method, an unsupervised machine learning algorithm, to group countries in clusters according to the diet of each country. Then they evaluated the average percentage of deaths per cluster and by analysing the data obtained they made several conclusions. Overall, the manuscript contains nice contributions. However, some minor issue should be addressed.

Authors { Answer#1 (A3.1)

Thank you to the reviewer for the nice words and the hard e_ort in improve the quality of our work.

Reviewer #3 Comment#2 (C3.2)

The title can lead to a false assumption. \Evaluation of Eating Habits in Relation to Deaths from COVID-19 Using Machine Learning Techniques" leads the reader to think that you used machine learning techniques to relate the eating habits with the death of COVID-19. You surely relate the two but not using machine learning techniques since you only used K-means clustering to group countries according to the diet of each country. In the main text page 4 line 140 you state this important information. So, I would slightly alter the title to not mislead the reader.

Authors { Answer#3 (A2.3)

We agree with the reviewer that the title can lead to a false assumption. We have changed it to Evaluation of Country Dietary Habits using Machine Learning Techniques in Relation to Deaths from COVID-19 which clearly summarizes the work done.

Reviewer #3 Comment#3 (C3.3)

Page 3 line 84: the mu and sigma are indicated as being the mean and standard deviation, but there is no indication of what. I suppose it is the mean and the standard deviation of the population. Page 3 line 85: the same comment as my previous one. I think it would be clearer if it is mentioned to what the mean refers to.

Authors { Answer#3 (A3.3)

As the reviewer has pointed out, in our work, the mean and standard deviation are related to the population. However, section 2.1 explains PCA globally without explicitly talking about the data set used. This is the reason why we do not indicate it in this part of the document.

Reviewer #3 Comment#4 (C3.4)

Page 3 equation 3: I think it would be clearer if there is a mention of which letter is used for the eigenvector.

Authors (Answer#5 (A3.4)

Thanks for the comment. We have already indicated the letter used for the eigenvector in the new version of the paper.

Reviewer #3 Comment#5 (C3.5)

Page 4 lines 141 and 142: It states \the percentage of deaths from COVID-19 for each cluster was evaluated" but there is no indication of how it was evaluated. This indication only appears on page 7 line 181: \the average percentage of deaths per cluster". I would be clearer if this indication appeared in this line.

Authors { Answer#5 (A3.5)

We fully agree with the reviewer on that point. It is not clear how the percentage of deaths is assessed for each group. We have added in this part of the document that this percentage is the average value for all the countries of the cluster by modifying the following lines: Once the countries were grouped together, the average percentage of deaths from COVID-19 for each cluster was evaluated to try to identify the dietary patterns that inuence the greatest number of deaths from detected infections.

Reviewer #3 Comment#6 (C3.6)

Page 6 line 166: It states \12 out of 30 countries appear in the top 30 of countries with more deaths… " but in Table 1 of page 7 there are 15 countries in green which \appear in the top 30 of countries with more deaths" and in page 7 line 191 it is referred that \46,67% of the countries that appear in the high deaths cluster

are among the 30 countries with the most deaths". This 46,67% correspond to 14 countries. So, it is not clear which is the real number of countries that appear in the top 30 of countries with more deaths.

Authors { Answer#6 (A3.6)

We are very grateful to the reviewer for detecting this typo. The correct results are shown in the table 1. For this reason, we have changed this line and also correct the percentage on the conclusion sections which is actually 50%.

Reviewer #3 Comment#7 (C3.7)

It would be a must to provide the code/scripts used in the analysis to make this work truly reproducible and to validate the results. The authors should add the code/script in the supplementary materials.

Authors { Answer#7 (A3.7)

Thanks for the comment. We have added the jupyter notebook with all the experiments in the supplementary materials to help other researchers to reproduce our experiments.